# Are Cry Studies Replicable? An Analysis of Participants, Procedures, and Methods Adopted and Reported in Studies of Infant Cries

**Giulio Gabrieli** [1] , **Giulia Scapin** [2] , **Marc H. Bornstein** [3] **and Gianluca Esposito** [1,2,*]

[1] Psychology Program, Nanyang Technological University, Singapore 639818, Singapore; giulio001@e.ntu.edu.sg

[2] Department of Psychology and Cognitive Science, University of Trento, 38068 Trento, Italy; scapin.giulia@gmail.com

[3] National Institute of Child Health and Human Development, Bethesda, MD 20892, USA; marc.h.bornstein@gmail.com

[*] Correspondence: gianluca.esposito@e.ntu.edu.sg

**Abstract:** Infant cry is evolutionarily, psychologically, and clinically significant. Over the last half century, several researchers and clinicians have investigated acoustical properties of infant cry for medical purposes. However, this literature suffers a lack of standardization in conducting and reporting cry-based studies. In this work, methodologies and procedures employed to analyze infant cry are reviewed and best practices for reporting studies are provided. First, available literatures on vocal and audio acoustic analysis are examined to identify critical aspects of participant information, data collection, methods, and data analysis. Then, 180 peer-reviewed research articles have been assessed to certify the presence of critical information. Results show a general lack of critical description. Researchers in the field of infant cry need to develop a consensual standard set of criteria to report experimental studies to ensure the validity of their methods and results.

**Keywords:** infant cry; acoustic analysis; cry analysis

## 1. Introduction

Cry is one of the first forms of communication newborns use to interact with their caregivers. Cry vocalizations are produced by the vibration of vocal folds that are controlled by the Central Nervous System (CNS). On this basis, researchers and clinicians have investigated the possibility of relying on acoustic analysis of infant cry to assess in a noninvasive way the integrity and developmental status of the CNS. In this regard, acoustic analysis has proven to be effective in identifying Autism Spectrum Disorder (ASD) [1–4], Sudden Infant Death Syndrome (SIDS) [5], and a variety of auditory-related problems during the early stages of development [6–10].

Historically, the first attempts at investigating acoustical properties of infant cry were conducted in the late 1960s. Since then, there has been exponentially increasing interest in the field, as demonstrated by the number of articles that have been published each year on the topic (Figure 1). In a typical cry study, vocal samples are collected from multiple infants who are induced to cry using a trigger, such as painful stimulus, while acoustic samples are recorded using one or more microphones and the signals are then stored in an analog or digital drive.

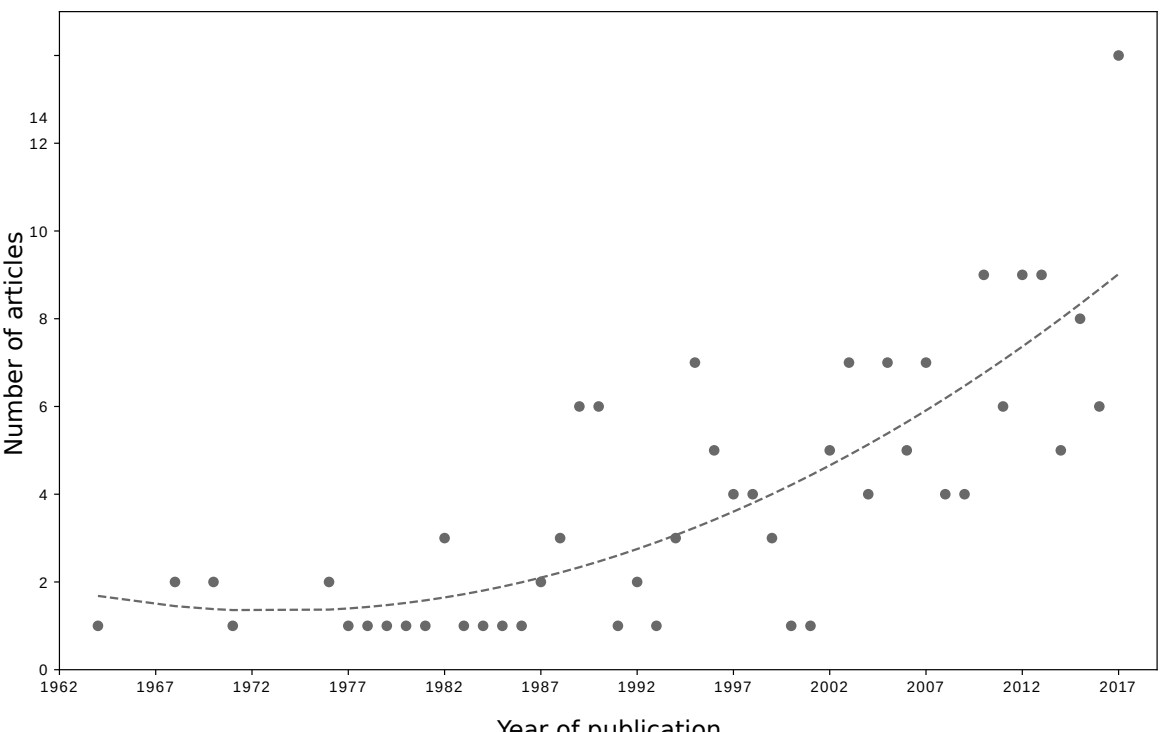

**Figure 1.** Number of articles per year of publication that succeeded our screening and met our inclusion criteria: Regression was evaluated using a 2nd-degree least square polynomial fit.

Acoustic analysis of infant cry is not usually based on the signal per se but on features extracted from the recordings [7,11]. These features are distinctive characteristics, such as the intensity of a specific frequency band. Among the most used features are the fundamental frequency (also defined as $F_0$) and its formants, the intensity of a frequency band, and duration and frequency of cry episodes.

Since the publication of the first monograph on cry analysis in 1968 [12], progress has been made in the hardware used for data collection, instrumentation employed for acoustic analysis, features of interest, and algorithms and methods used for data modeling. At first, researchers relied on visual inspection of cry spectrograms to extract meaningful information from cry samples. With the advent of more powerful computing technologies, interest shifted to methods that allow the estimation of acoustic features with higher accuracy.

Unfortunately, within the rich body of literature published on the acoustical properties of infant cry and its relationship with developmental pathology, it is not rare to find conflicting results. For example, differences in the acoustical properties of cry vocalizations of children at risk for ASD were found in Sheinkopf et al. [13] but not in Unwin et al. [14]. Likewise, the possibility of achieving a high level of accuracy in identifying deaf children by their cry vocalization was shown in Garcia and Garcia [15] but no differences between the properties of cry of deaf and normal-hearing infants were reported in Várallyay [16]. Different results can be attributed to common interindividual differences in cry vocalizations and, as Etz and colleagues pointed out [17], to a general lack of standardization in the field that precludes researchers from accurately replicating a given study. Specifically, no standardized guidelines exist that researchers can consult to report information about their participants, methods, data collection, and analytic procedures and no standardized datasets exist that can be marshaled to compare the performances of different statistical models to verify which is more effective in identifying cries of infants from a clinical population. Furthermore, according to Wermke and Mende, this lack of standardization seems generally accepted by the community of cry investigators [18].

Because of the nature of infants' vocalizations, results obtained by analysis of cry samples are defined and constrained by the pool of participants, instrumentation used, and methods employed

for feature extraction. In this article, we investigate how studies published in the field of infant cry have been conducted and reported, highlighting difficulties of replicating cry studies. We conclude with proposing a list of elements that researchers should consider when designing, conducting, and reporting studies of infant cry.

## 2. Materials and Method

### 2.1. Variable Definition

Two authors (G.G. and G.S.) assessed different acoustic manuals and publications published in the field of infant cry to identify the critical factors that affect cry studies at different stages of the experimental procedure. In a typical study of cry samples, three distinct categories of information can be distinguished: (1) participant information, (2) information about data collection processes, and (3) information about methods and data analysis, yielding 18 separate variables. Participation information is concerned with details about (a) the number of participants (Part), (b) the number of cry samples (Sam), (c) the age of the infants (Age), (d) the sex of the infants (Sex), (e) the trigger (Tri), (f) the position of the infants during the recording (Pos), and (g) the health status of participating infants (Hea). Regarding data collection, important are (a) the type of microphone used for data collection (Mic), (b) the Microphone-to-Mouth distance (MtM), (c) the recording environment (Env), (d) the sampling rate of recorded signal (SR), and (e) the file format used for storage (FF). Finally, with regard to the methods and data analysis, (a) the preprocessing procedure (PP), (b) the software and hardware used (SwHw), (c) the method of feature extraction (FE), (d) the frequency range analyzed (FR), (e) the features analyzed (AF), and (f) the window size of the signal during feature extraction (Ww) have all been investigated.

Detailed information about each variable is reported in Appendix A.

### 2.2. Search Methods and Results

To verify how information about participants, data collection, and methods and data analysis have been reported in published cry studies, we selected a set of articles following the preferred reporting items for systematic reviews and meta-analyses (PRISMA) [19]) guidelines.

An electronic database search on Elseviers' Scopus was conducted in March 2018, using the query *"Acoustic* Cry"*. This query was selected to identify all articles that contain any derivation of the stem acoustic and the word cry in the title or body of the article, and therefore, all articles where acoustical properties of cry were under investigation. Scopus returned 391 results. After automatic duplicate removal, 383 articles remained and were manually inspected for inclusion in the current review. One of the authors (G.G.) screened article titles and abstracts to exclude irrelevant articles. A total of 133 articles were excluded as irrelevant to this review. The remaining 250 articles were then examined by two independent coders to determine which articles met all four inclusion criteria, namely (1) analysis of samples were recorded from infants or from prerecorded datasets; (2) data collection procedure is explained or there is a clear reference to the employed dataset; (3) indications of the methodology used for signal processing is reported; and (4) the article was peer-reviewed and written in English. Seventy-seven articles were excluded because they did not meet the four inclusion criteria. A detailed representation of this screening procedure is given in Figure 2. The final dataset consists of 180 articles. Two independent coders (average between-coder agreement = 84%, evaluated on a subset of 20 randomly selected articles) who were blind to the aim of the review analyzed the included 180 and coded each for the presence of the 18 variables.

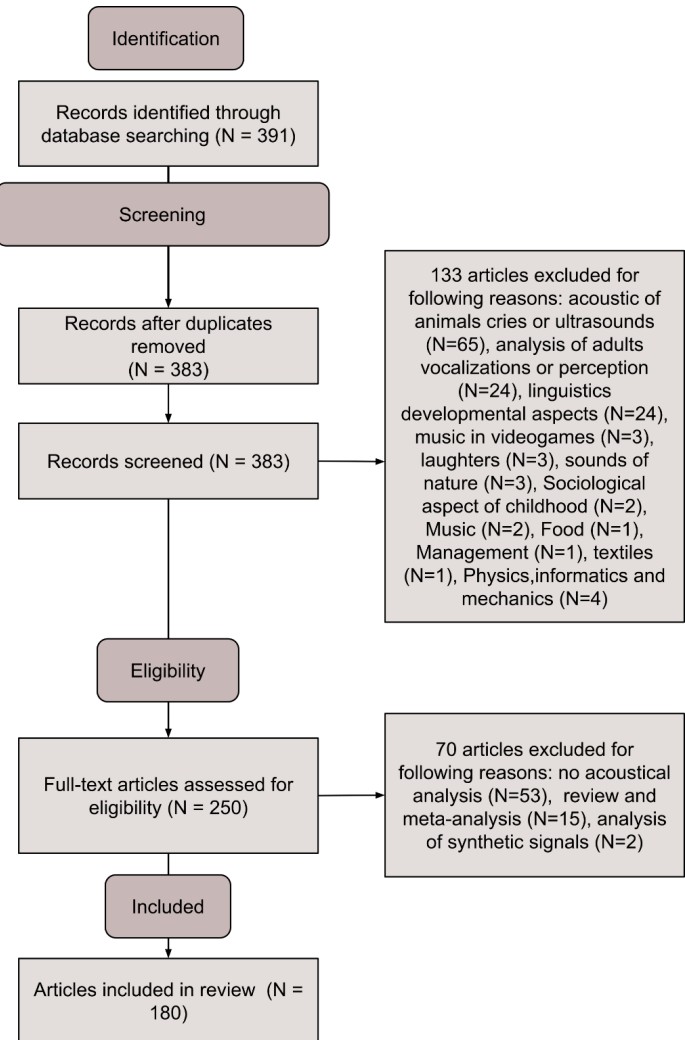

**Figure 2.** Article inclusion flow diagram (adapted from preferred reporting items for systematic reviews and meta-analyses (PRISMA) [19]).

## 3. Results

The percentage of presence of each variable is presented in Table 1. Results for each article can be found in the literature, and detailed notes for each manuscript are available in the online version of the dataset [20].

### 3.1. Participants

Overall, a very high proportion of the 180 articles reported (a) the number of participants in the experiment (N = 161, 89%) while (b) the number of cry samples analyzed in a study was reported in a slightly smaller number of articles (N = 122, 68%). Even though (c) the age of participants was reported in the majority of studies (N = 133, 74%), researchers appear less concerned in reporting (d) the gender of their participants (N = 71, 39%). The reason that induced a cry in infants, or (e) the Trigger, driven by the experimenter or spontaneous, was reported in many studies (N = 121, 67%). The (f) posture of the babies during the recording was reported in only a minority of studies (N = 55, 30%). Finally, studies that investigated infants suffering from pathological conditions almost all reported, when necessary, details about (g) the health status of their participants (N = 73 out of 77, 95%).

**Table 1.** Percentage of presence for each investigated variable (N = 180, where not differently stated).

|  | Variable | Abbr. | N | % |
|---|---|---|---|---|
| Participants | Number of Participants | Part | 161 | 89% |
|  | Number of Samples | Sam | 122 | 68% |
|  | Participants' Age | Age | 133 | 74% |
|  | Participants' Gender | Sex | 71 | 39% |
|  | Cry Trigger | Tri | 121 | 67% |
|  | Participants' Position | Pos | 55 | 30% |
|  | Participants' Health Status | Hea | 73 (77) | 95% |
| Data Collection | Microphones' Model | Mic | 112 | 62% |
|  | Mouth-To-Microphone Distance | MTM | 102 | 57% |
|  | Recording Environment | Env | 106 | 59% |
|  | Sampling Rate | SR | 115 | 64% |
|  | File Format | FF | 69 | 38% |
| Methods and Analysis | Preprocessing Procedure | PP | 98 | 54% |
|  | Software/Hardware | SwHw | 140 | 78% |
|  | Feature Extraction Methods | FE | 150 | 83% |
|  | Frequency Range | FR | 31 | 17% |
|  | Analyzed Features | AF | 161 | 89% |
|  | Windows Size | Ww | 53 | 29% |

There is great variability in the number of participants examined across these studies (One article may contain more than one study). They range from single-case studies (N = 5) to studies employing up to 1388 participants, with an average of 68.3 ± 150.7 participants per study. Similarly, the number of samples per study ranges from single-sample analysis (N = 1) to studies with up to 31,400 samples. The average number of samples per study is 729.1 ± 2861.5, with an average of 36.2 ± 221.4 samples per participant (evaluated on single-study reports where both the number of participants and number of samples were reported, N = 104). Ages of babies under examination in these papers range from 0 to 3 years of life, with the average participants aged 133.5 ± 181.1 days old. Sex of babies is more balanced for the number of males and female babies who participated in examined studies, with a total of 1709 females and 1663 males examined. Overall, a majority of the studies used pain-triggered vocalization (N = 76, e.g., heel prick), followed by spontaneous cries (N = 30), discomfort calls (N = 29), hunger-induced calls (N = 26), fear-induced calls (N = 30), and fussiness calls (N = 30). Additionally, in only 70 (39%) studies, the researchers relied on non-spontaneous vocalizations induced with a single trigger. For those studies, the situation is similar to the overall evaluation, with pain triggers being used most commonly (N = 50), followed by discomfort cries (N = 11), hunger (N = 8), and fussiness vocalizations (N = 1). More homogeneous is the position in which the babies were placed during the recordings, with the majority of the studies reporting that recordings were obtained with infants lying on their back (supine position, N = 42), with just a minority adopting a seating position (N = 14) and only one article in which infants were placed on their front (prone position, N = 1). Regarding the health status of babies, fourteen (N = 14) studies had a group composed by preterm babies, eleven (N = 11) had deaf infants, ten (N = 10) infants were diagnosed with or at risk of developing Autism Spectrum Disorder, and nine (N = 9) studies reported a group of infants suffering from asphyxia.

*3.2. Data Collection*

In describing their data collection procedures, more than half of the research articles reported (a) the model of the microphone employed during the recordings (N = 112, 62%) and (b) the mouth-to-microphone distance (N = 102, 57%). The (c) recording environment was clearly stated in almost six papers out of ten (N = 106, 59%), slightly fewer than (d) the sampling rate of recorded signals or of the speed of the recording tape (N = 115, 64%), but still fewer researchers indicated (e) the file format used for storage (N = 69, 38%), as tape, disk, or digital.

For the model of microphone, researchers relied on many different models and manufacturers (for a complete list, please refer to Esposito and Gabrieli [20]), but much more consistency is present for

the most adopted mouth-to-microphone distance, where a 0.15-m distance has been used in thirty-five (N = 35) studies, followed by 0.20 m (N = 13) and 0.30 m (N = 11). An unbalanced situation is presented for the recording location, where studies employing recording collected in clinical situations (N = 75) almost double the number collected in nonclinical settings (N = 40). Only a minority of studies (N = 8) employed cries collected in both a clinical and nonclinical setting. As for the sample rate of recorded signals, a vast majority of collected samples were sampled at 44,100 Hz (N = 30), 10,000 Hz (N = 18), 16,000 Hz (N = 12), and 48,000 Hz (N = 13), with signals mostly recorded on magnetic tapes (N = 52) and in WAV (Waveform Audio File Format) format (N = 15).

*3.3. Methods and Data Analysis*

Amongst those 180 studies, about half (N = 98, 54%) reported (a) the preprocessing procedure, when one was applied. A majority of examined articles provided information about (b) the software or hardware used for the analysis (N = 140, 78%) and (c) the feature extraction methods (N = 150, 83%). However, only a small number clearly stated (d) the region of interest within the spectrum (N = 31, 17%). (e) Studied features were clearly listed in almost all articles (N = 161, 89%), but (f) the window size used during the analysis was reported in only less than a third of articles (N = 53, 29%).

Overall, a manual or automatic segmentation of signals is usual(N = 54), preceded or followed by filtering (N = 44). In various papers, only a subset of extracted segments (for example, only the first three cry vocalizations of each participant [21]) was used in subsequent analysis (N = 32). As for the software used for analysis, KAY's Computer Speech Lab was the most used tool (N = 24) until recently. It has since been slowly replaced by Praat (N = 36). However, Matlab remains researchers' favorite scripting language (N = 31). Despite the effort in reporting the used software and instrumentation, the software version or model of hardware was only reported in fewer than half of the cases (N = 107 out of 246, 43%). This is also reflected in feature extraction methods, in which methods based on the analysis of the spectrum of an audio sample dominate (FFT N = 48, MFCC N = 22, and LTAS N = 15). Concerning frequency range, the majority of studies focused on frequencies below 10,000 Hz (N = 32), with a great interest in frequencies up to 5000 HZ (N = 21). The majority of studies analyzed the fundamental frequency ($F_0$, N = 124) of cry samples, the duration of cry vocalizations (N = 82), and the energy conveyed by the signal (N = 60) as some of the most investigated acoustic features. Windows ranged from 5 ms to 290 ms, with 25 ms (N = 20) and 50 ms (N = 14) being the most usual window sizes.

## 4. Discussion

Results of our analysis confirm Etz and colleagues' conclusion about the general lack of standardization in reporting infant cry analysis studies.

When it comes to the information about the infants participants whose vocalizations have been reported and analyzed, researchers usually stress the number of babies who took part in the experiment as well as the number of recorded samples, age, cry trigger, and health status. Little stress is placed on the importance of reporting participants' gender and position assumed during the recording. Taken together, this profile of information raises implications about the reproducibility of infant cry studies. If it is true that the gender of a babies affects the properties of their vocalizations [22–24], when this information is not reported, researchers will not be able to assume that acoustic properties of infants' vocalizations are normally distributed or that deviations from normality are not present because of unbalances in the number of male and females babies being compared.

Similarly, when researchers report the experimental setup they employed for their studies, in just about six out of ten articles, the model of microphone used to record signals, the distance between the babies' mouth and the microphone, the sampling rate of recorded signal, or the recording location have been reported, and the file format used for storing recorded signals has been reported in even fewer cases. The problem with such missing information is that, even in cases where the same microphone is used across different studies, researchers are unable to ensure that recorded signals are comparable if microphones were placed at different distances, recorded in different locations, or with different

sampling rates. Especially relevant for digital signals, the format used for storage plays an important role in the preservation of recorded frequency, as different compression algorithms may alter the frequency information conveyed in infants' vocalizations (additional details on file compression are reported in Appendix A.2.5).

Finally, regarding the methods and data analysis sections of their papers, researchers are consistent in reporting the software and hardware used in their work (even though the software version or hardware version was reported only in approximately half of the cases studied), the feature extraction procedure, and the list of features used. Less agreement is found in indicating the preprocessing procedure that the signal had undergone prior to the analysis, the region of interest within the spectrum, and the window widths used to study those features. However, knowledge about software version is crucial, especially for feature estimation and the parameters used to process the signals. For example, among the 180 articles we investigated, although Praat was widely adopted by different research groups, only 17 cases reported the software version used (out of the 36 versions in total which have been employed), even though more than four hundred different releases of Praat are available (https://github.com/praat/praat/releases, 416 releases as per 5 November 2019). Additionally, different preprocessing procedures may affect results obtained by different studies, and therefore, a clear indication of the steps adopted in a study should always be reported. At the same time, we acknowledge that current methods of reporting analysis may change as more and more scientists embrace open practices, such as data, script, and software sharing.

One of the limitations of infant cry studies is the absence of standardized datasets to compare novel approaches with traditionally employed techniques. It is not uncommon to see that researchers collected data for their works that were not published with the manuscript or in external repositories or that they employed outdated datasets, such as the Baby Chillanto Infant Dataset of which the web page is no longer available [25].

With this work, we hope to promote a constructive discussion on how to standardize current and future investigations among cry researchers to enhance the reproducibility of investigations of infant cry and to facilitate the adoption of cry-based technologies for clinical purposes. Future publications in the field should contain all the information needed to allow a critical interpretation of results based on participants' demographic information, data collection procedures, and methods and data analysis and to accurately replicate the study. Moreover, researchers should, whenever possible, share their original recordings. Accessible datasets may positively impact the quality and quantity of published research through the development of new and improved methodologies for feature extraction, which can then be bused in a clinical environment for early diagnosis of developmental atypicality.

## 5. Conclusions

This study investigates how cry research has been conducted and reported in the last half century. In 1995, Robb [26] wrote that the absence of acoustic validation studies was surprising for two reasons: first, there are researchers who state that acoustical analysis of cry is diagnostically significant and, second, the unique configuration of an infant's vocal tract is difficult to ignore. Unfortunately, after more than 20 years, the presence of one or more standardized datasets and guidelines for analysis and publication is still missing.

In this review, 180 research articles are analyzed for the presence of variables that can be used to replicate, compare, and make assumptions about relations between acoustical features of infant cry and infant developmental status. We found a pervasive lack of critical description regarding various aspects of samples and their properties, processes of data collection, and methods and data analysis. This dearth of information is accentuated by the paucity of freely available datasets to test and compare feature extraction methods. Similarly, to the best of our knowledge, there are no guidelines for reporting variables to effectively explain results obtained in cry analyses. Researchers in the field should develop standardized ways to report experimental studies to ensure the validity and reproducibility of their methods and results. In conducting cry-based research, researchers should aim to obtain multiple

samples from multiple infants to minimize the impact of single samples and single subjects on the final results. Additionally, age and gender of samples should be balanced or minimized to avoid effects of those variables on final inferences (when the two are not the independent variable under investigation). Moreover, to avoid differences due to the type of vocalizations, researchers should rely on non-spontaneous vocalizations induced with a single trigger and they should avoid recording babies placed in different positions. More freedom should be given to the selection of the recording environment, type of microphone, the distance of the microphone from the infants' mouth and sampling rate, given that a proper rationale is given and that all the details are reported in the final publication. For the file format, we suggest using lossless compression file types in order to preserve the acoustical properties of recorded vocalizations. Finally, the analysis process, including the preprocessing (if any) and feature extraction procedures and software used (with versions and employed settings) should be clearly reported. Our proposed checklist for the reporting of cry studies information about participants, data collection, and methods and analysis is reported in Appendix B. We hope that this discussion inspires self-evaluation of participant selection, data collection, and subsequent methods and data analysis. Availability of reliable results affects the ability of pediatricians to recognize pathology and developmental problems in their early stages using noninvasive techniques and generates discussion among researchers for the best methodologies to apply when analysing infant cry.

**Author Contributions:** Conceptualization, G.G., G.S., M.H.B., and G.E.; methodology, G.G., G.S., and G.E.; formal analysis, G.G.; data curation, G.G. and G.E.; writing—original draft preparation, G.G.; writing—review and editing, G.G., G.S., M.H.B., and G.E.; visualization, G.G.; supervision, G.E.

**Funding:** This research was supported by Nanyang Technological University (Singapore) under the NAP-SUG grant; the Intramural Research Program of the NIH/NICHD, USA; and an International Research Fellowship at the Institute for Fiscal Studies (IFS), London, UK, funded by the European Research Council (ERC) under the Horizon 2020 research and innovation programme (grant agreement No 695300-HKADeC-ERC-2015-AdG).

**Acknowledgments:** The authors thank to Anais Ang Xin Hui, Chiara Iannaccone, Giulia Garbin, Mengyu Lim, and Pamela Goh Pei Lin for their help.

**Conflicts of Interest:** The authors declare no conflict of interest.

## Abbreviations

The following abbreviations are used in this manuscript:

| | |
|---|---|
| CNS | Central Nervous System |
| ASD | Autism Spectrum Disorder |
| SIDS | Sudden Infant Death Syndrome |
| $F_0$ | Fundamental Frequency |
| $F_n$ | nth Formant |
| Part | Number of Participants |
| Sam | Number of Samples |
| Age | Participants' Age |
| Sex | Participants' Gender |
| Tri | Cry Trigger |
| Pos | Participants' Position |
| Hea | Participants' Health Status |
| Mic | Microphones' Model |
| MTM | Mouth-To-Microphone Distance |
| Env | Recording Environment |
| SR | Sampling Rate |
| FF | File Format |
| PP | Preprocessing Procedure |
| SwHw | Software / Hardware |

FE      Feature Extraction Methods
FR      Frequency Range
AF      Analyzed Features
Ww      Windows Size

## Appendix A. Supplementary Material—Variable Descriptions

### *Appendix A.1. Participant Information*

Key variables associated with participant information include the number of participants (Part), number of cry samples (Sam), age (Age) and sex (Sex) of the infants, trigger (Tri), the position of the infants during recording (Pos), and infant health status (Hea).

### Appendix A.1.1. Number of Participants (Part)

The number of participants investigated in a cry study is an important parameter to consider for several reasons. Cry vocalizations are highly variable both within samples collected from the same subject and between the subjects. Due to between-subject variability, results based on small numbers of participants may induce errors in interpretation [27,28]. For example, the application of predictive models based on training done on a small number of participants results in models that are not suitable for generalization to a wider population [29].

### Appendix A.1.2. Number of Cry Samples (Sam)

Because of the high within-subject variability [17], the number of different cry samples collected in a study can be used to assess the extent to which obtained results are influenced by a single individual's cry recordings. This shortcoming holds especially for studies with limited numbers of participants and few cry samples for each. In these cases, the properties of a single sample can severely influence the average values of the whole dataset. Studies with large numbers of samples collected from a small pool of participants may be affected by overfitting, with statistical models becoming too sensitive to the investigated individuals.

### Appendix A.1.3. Age of the Infants (Age)

As cry is produced by the vibration of the vocal folds that grow and change with an infants' age, acoustical properties of cry are influenced by the age of investigated participants [30]. Some authors suggest reporting gestational age at birth as well, together with infants' weight both at birth and during data collection [22,31].

### Appendix A.1.4. Sex of the Infants (Sex)

Controversial results on cry have been reported regarding the sex of infants. Multiple studies (e.g., Sahin et al. [22], Borysiak et al. [23], and Goberman and Whitfield [24]) found significant differences between the acoustical properties of cry vocalizations of girls and boys, while in other studies (e.g., Fuller and Horii [32]) no differences were found or reported.

### Appendix A.1.5. Trigger (Tri)

Acoustical properties of cry reflect the reason that induced the cry [33,34]. Vocalizations obtained from different functional roles (e.g., pain or hunger) are not comparable because cries from different trigger categories convey different frequency information. Researchers should report the trigger used to induce babies to cry and compare only cries obtained using the same trigger.

### Appendix A.1.6. Position of the Infant during Recording (Pos)

Infants' body position during cry recording has been found to influence the acoustic properties of the cry. This influence is separate from the developmental status of the infant. For example,

Goberman et al. [35], identified differences in cry acoustics of infants recorded in a supine versus prone position even in response to the same pain stimulus [36,37].

Appendix A.1.7. Health Status of the Infants (Hea)

Infants' health and developmental statuses are reflected in the acoustic properties of cry. This peculiarity is the basis for research on early screening of pathological problems. Specific differences in cry features are associated with different pathologies. For example, a higher fundamental frequency in cry utterances is associated with a higher risk of autism spectrum disorder diagnosis [30,38–40], while hearing-impaired infants produce longer vocalizations with lower second formant ($F_2$) and less energy in the higher frequency bands [41,42]. Knowledge about the health status of investigated infants is necessary to correctly evaluate the obtained results.

Appendix A.1.8. Additional Information

Additional information should be reported to clarify the demographics of both the infants and their caregivers. Different studies have, in fact, identified a relationship between caregivers' spoken languages and the acoustic vocalizations of the infants [43,44]. Authors should consider reporting any other information about the demographics of the participants that may help to understand and give context to obtain results (e.g., How were the participants recruited? What is their ethnic background? How old are the infants' caregivers?).

*Appendix A.2. Data Collection*

During data collection, several aspects of the experimental setup influence the quality and properties of recorded signals. We identified six key variables: characteristics of the microphone used for data collection (Mic), the microphone-to-mouth distance (MtM), recording environment (Env), the sampling rate of recorded signal (SR), file format used for storage (FF), and number of channels employed for data recording (NC).

Appendix A.2.1. Microphone Used for Data Collection (Mic)

The type and model of microphone used are necessary to ensure that data collected using the described experimental setup is suitable for the analysis of investigated frequencies. Different microphones respond optimally to specific frequency ranges, according to the type of technology employed and directionality of the microphones [45]. The directionality of a microphone, expressed as a polar pattern, indicates its sensitivity to sounds coming from different directions. Omnidirectional microphones respond in the same way to sound waves coming from different directions; cardioid microphones are sensitive to sound waves coming from a specific direction, limiting the external noise coming from other directions [46] (Chapter 3). Future studies may also address the possibility of using contact microphones to investigate directly the activity of the infants' vocal folds. Microphone sensitivity is represented using a frequency response chart, which graphically represents microphone response (in dB) to each frequency with the source at a specific distance from the microphone. An example of a frequency response chart appears in Figure A1. Microphones employed in cry studies should have a homogeneous response to the investigated frequencies that are based on the features of interests and methodology employed for extraction. For example, for a direct estimation of the fundamental frequency ($F_0$), sensitivity should be homogeneous in frequencies below 1 kHz, while for indirect estimation of $F_0$ from the first four formant peaks, the response should be homogeneous from 700 Hz and 3 kHz.

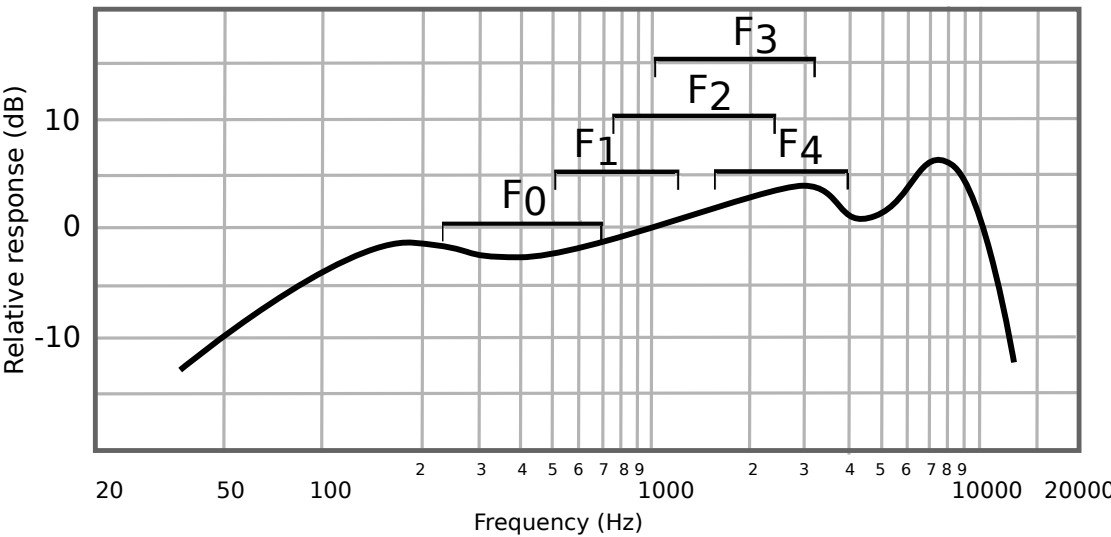

**Figure A1.** Example of microphone response chart: Infant cry region of interests for extraction of the fundamentals and the first four formants are highlighted. Usually, frequency response charts are recorded at a specific Microphone-to-mouth (MtM) distance.

Appendix A.2.2. Microphone-to-Mouth Distance (Mtm)

Microphones have specific frequency responses and polarity. Microphone distance to the infant's mouth influences the intensity of recorded signals as well as the intensity of external noises, such as wind [46] (Chapter 10). In free space, a sound is propagated uniformly in every direction. According to the inverse square law, the intensity of sound in a free field is inversely proportional to the square of the distance from the source [47] (Chapter 1). Different microphones have different optimal distances, at which the response to a specific frequency range is flat. For example, vocal microphones are designed to produce a flat frequency response at 5 to 10 cm from the mouth, while noise-canceling microphones are most effective when positioned next to the mouth [46] (Chapter 17). To maximize the intensity of cry, to minimize external noise, and to obtain a flat response in the frequency range of interest, microphone-to-mouth distance has to be carefully evaluated.

Appendix A.2.3. Recording Environment (Env)

Laboratory studies have samples recorded within the same environment, removing possible variance in the recordings arising from the different acoustics of different rooms. Acoustic characteristics of recordings obtained in different places, such as at different infants' homes, are influenced by nonidentical wave reflection, which produces shifts in recorded frequencies [47] (Chapter 7). These reflections are modulated by the distances between microphones, infants' mouths, and walls, as well as by the walls' absorption coefficient [48] (Chapter 2). When recordings are done in home settings, there may be the presence of additional and noncontrollable background noise as well as of the caregivers' voices. The cry trigger also modulates the acoustical properties of recorded signals. Ross et al. [49] investigated differences between distress situations in 12- to 18-month-old infants at home or in laboratory conditions. When tested in unfamiliar environments, children cried almost three times as long as when they have been tested at home. Furthermore, in laboratory studies, the presence of others has to be taken into account, especially where soundproofed rooms are not available. Infants cry when hearing cries of other infants [50–52]. In hospital situations, where many infants are present, tested infants may be able to hear cries of many other infants or patients. To avoid empathic distress responses, it is important to prevent tested infants from hearing the cry vocalizations of other individuals. Additionally, future studies should consider reporting other information about the

environment that may play a role in infants' vocalization, such as the temperature of the room and the humidity of the air.

Appendix A.2.4. Sampling Rate of Recorded Signal (Sr)

The sampling rate (SR) is the number of samples per second contained in a digitalized continuous signal [53]. A signal sampling rate influences the highest detectable frequency and frequency resolution as well as the accuracy of extracted features. The maximum detectable frequency, or Nyquist frequency, corresponds to half of the sampling frequency [54,55]. The extraction of acoustic properties in the frequency domain is influenced by the signal's resolution, which is the smallest detectable change within subsequent points of a signal. For audio signals, the frequency resolution after applications of a Discrete Fourier Transform (DFT) is given by the ratio between the sampling rate of the signal and the number of points used in the DFT. During data processing and feature extraction, variation in the sampling rate results in distortion of the original signal and, therefore, in the frequency resolution and highest detectable frequencies and in the accuracy of extracted frequency information.

Appendix A.2.5. File Format Used for Storage (Ff)

Cry data can be stored as analog or digital forms. Today, the majority of researchers opt for direct storage on digital devices or analog-to-digital conversion. When stored on a digital drive, digital files are encoded in a specific file format (FF). Different file formats store files according to different acoustic properties, and therefore, the original signal is adapted when stored. There are three different types of audio file formats: uncompressed, lossless compressed, and lossy compressed [56] (Chapter 5, p. 157ff).

- Uncompressed files store the signal as it is, applying no content compression and resulting in files taking more space on digital drives.
- The lossless compressed format encodes in a way that reduces the size of an input file by creating a copy with the same acoustical properties that may have a smaller size, usually in the ratio 2:1 [57].
- To achieve a greater reduction in file space, a lossy compression algorithm can be used. Lossy compression achieves a higher compression ratio, usually around the ratio of 10:1, by reducing the audio quality of the signal. Although quality loss is almost imperceptible to human ears, modification of original signal influences the quality and accuracy of acoustical features estimated from it, such as $F_0$. The most popular lossy file format is the MP3 format, which is widely used for music compression, but it is also employed in the research environment.

Moreover, the number of bits used for recording allows getting a more precise digital representation of each recorded sample. This enhanced representation allows having a higher difference between the lowest and highest sounds that can be recorded, such as the higher the number of bits, the higher the difference. Knowledge about the number of used bits may allow comparing results obtained in different studies in a more accurate way as well as understating the rationale behind specific preprocessing procedures [58].

*Appendix A.3. Methods and Data Analysis Information*

We identified six key variables that influence accuracy and precision of preprocessing techniques and data analysis results: the preprocessing procedure (PP), software and hardware used (SwHw), feature extraction method (FE), analyzed frequency range (FR), analyzed features (AF), and window size of the signal during feature extraction (Ww).

Appendix A.3.1. Preprocessing Procedure (Pp)

Once collected, if data contain external noise or artifacts, there may be a manual or automatic screening of "good" versus "bad" samples, with the latter removed before the analysis. Before the data analysis, recorded signals may undergo a series of treatments to increase the signal-to-noise ratio and to enhance the accuracy and robustness of extracted features. Preprocessing can modify signals

in different ways. For example, signals can be parsed for noise removal and segmented into smaller fragments or downsampled. Downsampling a signal to reduce computational time or to save space on hard drives results in a reduction of information stored in a signal. This reduction leads to shifts in frequency information [59]. Similarly, the application of digital filters to increase the signal-to-noise ratio may alter the properties of investigated frequencies (FR) [60] (Chapter 1). For this reason, preprocessing procedures (Pp) have to be clearly reported by providing all the information required to correctly evaluate or replicate the methodology, stating all the possible alterations of a signal.

Appendix A.3.2. Software and Hardware (Swhw)

With the advent of computer-based methodologies, more objective and accurate digital analyses of quantitative acoustic parameters are now available [61]. In all investigations, the original cry signal underwent modification because of the hardware used for data collection and methodologies applied during preprocessing and data analysis. During preprocessing and subsequent analysis, researchers use different software and instrumentation (SwHw). Software name, version, and parameters need to be specified as new releases often correct previous bugs within the code that may have generated incorrect analyses. Furthermore, changes in customizable parameters in different software lead to differences in the accuracy and precision of extracted features. To estimate the fundamental frequency of cry samples, researchers often use Praat, an open-source software designed for voice analysis. Praat's source code repository (a metadata container) received more than 2400 commits (a change in one or more files) and 390 different complete releases of the software (as of 6 April 2018). Significantly, Praat's default frequency range, from 75 to 500 Hz, is not suitable for an accurate analysis of infant cry because healthy infant cries vary over a frequency range of 300 Hz to 600 Hz or higher for infants with developmental pathologies, such as ASD [62–64].

Appendix A.3.3. Feature Extraction Method (Fe)

Several methodologies can be employed to estimate the value of $F_0$ from a cry sample. For example, it can be done by direct estimation from peaks in the investigated frequency range, between about 200 to 700 Hz, or by regression from formant peaks, averaging the ratios between formant peak frequencies and their order [65,66]. Those methodologies require low levels of computations, but on the downside, their robustness to noise, especially in the frequency bands of interests, is very low. Another class of methodologies widely accepted by researchers is the estimation of $F_0$ and its formants using the cepstrum approach. The cepstrum is defined as the inverse discrete Fourier transformation (DFT) of the logarithmic magnitude of the DFT of a signal, causing a compression of the dynamic range and reducing amplitude differences in the formants. Algorithms based on cepstrum can separate coefficients associated with the glottal excitation and the vocal tract and proved to be suitable for the analysis of both the adult voice and the infant cry [67]. To analyze recorded data using algorithms based on the cepstrum, preprocessing on the signal is necessary to ensure the analysis of a clean signal, and therefore, a preprocessing stage is required. Since the 1960s, methodologies employed in feature extraction evolved with the development of new technologies. While initially features were estimated manually by reading the spectrogram of a signal, the advent of computers enabled the development of automatic and semiautomatic methods for feature extraction. Computer-based methodologies chain together several stages, performed one after the other. During those stages, the original signal undergoes a series of modifications that are reflected in obtained results (see for example Boersma [64], where all steps of an algorithm are provided and explained in detail).

Appendix A.3.4. Analysed Frequency Range (Fr)

During data analysis, researchers focus on a specific frequency range (FR) of the spectrum, for example, by digitally filtering the signals or by selecting only a subset of frequency bins after passage to the frequency domain. The selection of a specific frequency range avoids low-frequency noise and interaction between higher frequencies. In Praat, for example, it is possible to specify

a frequency range in which to search for $F_0$. As introduced in the paragraphs above, Praat's default settings are aimed at $F_0$ estimation in the field of voice analysis and therefore, are not suitable for its estimation on infant cry samples [64]. Details about specified parameters of commercial software or self-developed tools give a better understanding of the accuracy of extracted features.

Appendix A.3.5. Analyzed Features (Af)

Analysis of infant cry can be done by using different features, both in the time and frequency domains. Tahon and Devillers [68] investigated acoustic features for emotion recognition and identified 174 different features, in time and frequency domains. Providing a list of analyzed features ensures that others can evaluate—based on investigated sample, experimental setup, and methodologies employed—the accuracy of obtained results.

Appendix A.3.6. Window Size of the Signal during Feature Extraction (Ww)

During feature extraction, algorithms are applied to smaller portions of the original signal, called a window. As signals are composed of waves at different frequencies, feature extraction works by investigating the repeating patterns within those windows. The size of the windows (Ww), as well as the overlap and distance (step size) between subsequent windows, affects the resolution of estimated features. Translation of the signal from the time domain to the frequency domain is done by dividing the spectrum into frequency bins. The width of the bins is given by the ratio between the sampling frequency of the signal and the number of time points used in the Fast Fourier Transform [69] (Chapter 3). The finer a window is, the higher is the number of possible consecutive analyzable time points, but the frequency resolution of the signal is reduced. To overcome this problem, it is possible to use longer windows (higher frequency resolution), by overlapping subsequent windows (in order to maintain a high number of analyzable time points). When employed, the window size, step size, and overlapping should be reported in order to allow for correct replication of employed methods.

**Appendix B. Supplementary Material—Checklist**

*Appendix B.1. Participants' Information*

☐  Number of participants: expressed as total number of participants of the study and with clear indication of the number of participants per group (if more than one group is present).

☐  Number of samples: expressed as total number of samples recorded and with clear indication the number of samples per group (if more than one group is present) and of the number of samples per participant.

☐  Age of the participants: statistics (mean, std, min, and max) age of the participants of the study for the whole set of participants and for the subset of participants per group (if more than one group is present). If possible, researchers should also indicate the gestational age at birth. Whenever possible, the weight of the participants should be reported as well.

☐  Gender of the participants: total number of male and female participants and reported per group (if more than one group is present).

☐  Cry trigger: information about the trigger that has been used to induce crying vocalizations in babies.

☐  Posture during the recording: information about the position of the babies during the recordings (supine, prone, and seated).

☐  Additional information: any other additional information that may help giving context to obtained results (e.g., language, ethnicity, and recruitment process).

*Appendix B.2. Data Collection*

☐    Microphone model: the model of the microphone(s) used for recording.

☐    Mouth-to-microphone distance: distance between the infants' mouths and the microphone.

☐    Recording environment: environment in which the data have been recorded (clinical or nonclinical). Additional information (e.g., was the baby familiar with the environment? Was the room soundproof and or silent? Was the temperature in room controlled? Was the level of humidity in the room controlled?) should be reported to clarify where data have been collected.

☐    Sampling rate: Sampling rate of recorded signal (and resolution in bit).

☐    File Format: format in which the file has been saved.

*Appendix B.3. Methods and Analysis*

☐    Preprocessing procedure: detailed information about the preprocessing steps should be reported, included settings and parameters of employed tools and software.

☐    Software and hardware: information about the software (with versions) and hardware (with model) employed in the research.

☐    Feature extraction procedure: procedures that have been used to estimate analyzed features (if necessary).

☐    Region of interest: frequency regions of interest of the signals that have been processed (e.g., between 100 and 4000 Hz).

☐    Investigated features: list of features that have been analyzed.

☐    Window size: size of the windows employed in the study, if any, including overlapping and step size.

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
