# Peer review of "Are Cry Studies Replicable? An Analysis of Participants, Procedures, and Methods Adopted and Reported in Studies of Infant Cries"

_acoustics, doi:10.3390/acoustics1040052_

Round 1

Reviewer 1 Report

The authors of this manuscript provide a review of 180 studies over the last 60 years on acoustic infant cry analysis. The review focusses on predefined variables related to reported participant information, information on the data collection process, and information on methods and data analysis. Results point to a general lack of reported information required for the reproducibility of and the comparability between cry studies. The paper is well written and represents a valuable and highly relevant contribution not only for the “infant cry community”, but for all research communities building upon recorded audio signals, urging towards a higher degree of standardisation and transparency in both study procedures and dissemination. However, from my point of view there are a few minor issues with regard to CONTENTS and FORM to be addressed prior to publication:

CONTENTS

- At the end of the paragraph “Acoustic analysis of infant cry is usually not based on …” in the Introduction section, the authors should give some references.

- At the beginning of Section 2.1., the authors should provide a more detailed description of the variable definition process. Who was responsible for defining the three variable categories and the corresponding sub-categories? On the basis of which criteria was this done? In the Abstract, the authors state that “available literature” was used to identify these categories. However, in the Methods section of the manuscript, no information is given on this step.

- In the Checklist in Appedix B, the authors suggest also to report the recorded signal’s amplitude resolution in bit. Why has the bit depth – which also influences subsequent feature extraction processes – not been defined and consequently analysed as sub-category of data collection in this review?

- As feature extraction is often applied on the basis of overlapping time windows, the authors should also report on the variable “overlap” or “step size” in addition to the variable “window size”. In case of overlapping time windows, not the window size, but the step size determines the time resolution of the extracted feature contour. In the last sentence of Appendix A.3.6., the author’s should add the word “frequency” before “resolution” (… but the frequency resolution of the signal is reduced.).

- In Section 2.2., the authors describe that articles not meeting all four inclusion criteria were excluded. However, the four inclusion criteria were not reported. The authors should add them in this section.

- With regard to the literature search procedure, I was confused about the fact that even though only the search term “Acoustic* Cry” was used, also papers with variations of “cry” in the paper title were found, namely “crying” and “cries”. How can this be explained? Moreover, I wondered if more relevant articles would have been found, if specific acoustic terms would have been additionally used as search queries, such as “fundamental frequency”, “spectr*”, “cepstr*”, ...

- Section 3.1.: The authors state that the age of participants in cry studies ranges from 1 to 4 years with an average age of 133.5 days. Do the authors really mean 1 to 4 years rather than 0 to 3 years, i.e. the first to the fourth year of life?

Could the authors please also present results on which single triggers were used for cry induction in the 70 studies that relied on non-spontaneous vocalisations?

- At Checkbox 2 on Participant information in Appendix B, the authors should add that also the number of samples per participant has ideally to be reported in order to avoid that a group is misinterpreted by reason of a high number of samples by a single participant within the group.

- At the Checkbox on “Region of Interest” in Appendix B, the authors should add the term “frequency” before “region of interest” in the explanation (frequency region of interest of the signal …).

- Appendix A.2.3.: The authors should add that voice samples of interest recorded in home settings are often overlaid by background noise, such as everyday noise sources or parental voice.

- Appendix A.2.4.: How far does the sampling rate of a signal influence the lowest detectable frequency?

- The authors should specify (in the Figure legend) the source-microphone distance used for measuring the microphone frequency response shown in Figure A1 (usually 1m?).

FORM

- Conclusion: In case, Robb 1995 was quoted literally (“firstly, there are researchers who …”), the authors should use quotation marks.

- What does “Tx” at the end of the second bullet point in Appendix A.2.5. mean?

- In the last sentence of the Appendix A.3.5., the authors should use an en dash surrounded by blanks between “evaluate” and “based”, and between “employed” and “the accuracy”.

- The authors should not mix up American and British English (“standardization” vs. “standardised”).

- In the reference list, the authors should not mix up upper case and lower case formatting styles for journal names (e.g., “Journal of Voice” vs. “Clinics in developmental medicine”).

- Both the second and the third Sub-Checklist headers in Appendix B are entitled “Data Collection”.

- There are some spelling/grammar errors, inconsistencies, and sentence constructions that hamper readability and text understanding, e.g., “analogical or digital” (analog), “the format used for storage play an important role …” (plays), “number of male and females babies …” (female), “a open-source software” (an), “signal to noise ratio” vs. “signal-to-noise ratio” (the authors should use the second notation), “feature extraction” vs. “features extraction” (the authors should consistently use the first term), “over a frequency range 300 Hz” (of), “Acknowledgments: the authors are thankful” (The), “the the number”, “Feature Extraction Procedure: procedures has been used to estimated analyzed features” (procedures that have been used to estimate), ...

Author Response

CONTENTS

- At the end of the paragraph “Acoustic analysis of infant cry is usually not based on …” in the Introduction section, the authors should give some references.

Thanks for pointing this out. References were added to support this sentence. (Line 28)

LaGasse, L. L., Neal, A. R., & Lester, B. M. (2005). Assessment of infant cry: acoustic cry analysis and parental perception. Mental retardation and developmental disabilities research reviews, 11(1), 83-93.

Esposito, G., Nakazawa, J., Venuti, P., & Bornstein, M. H. (2013). Componential deconstruction of infant distress vocalizations via tree-based models: A study of cry in autism spectrum disorder and typical development. Research in Developmental Disabilities, 34(9), 2717-2724.

- At the beginning of Section 2.1., the authors should provide a more detailed description of the variable definition process. Who was responsible for defining the three variable categories and the corresponding sub-categories? On the basis of which criteria was this done? In the Abstract, the authors state that “available literature” was used to identify these categories. However, in the Methods section of the manuscript, no information is given on this step.

Thanks for highlighting this. The rationale behind the variables selection and the authors in charge for this have been added (Line 59)

- In the Checklist in Appedix B, the authors suggest also to report the recorded signal’s amplitude resolution in bit. Why has the bit depth – which also influences subsequent feature extraction processes – not been defined and consequently analysed as sub-category of data collection in this review?

Thanks for pointing this out. We added a note on why the resolution is important in A.2.5 (Line 416)

- As feature extraction is often applied on the basis of overlapping time windows, the authors should also report on the variable “overlap” or “step size” in addition to the variable “window size”. In case of overlapping time windows, not the window size, but the step size determines the time resolution of the extracted feature contour. In the last sentence of Appendix A.3.6., the author’s should add the word “frequency” before “resolution” (… but the frequency resolution of the signal is reduced.).

Thanks for highlighting the typo. We agree that overlap and step size should be reported and we have therefore added a paragraph in both Appendix A.3.6 and B. 

- In Section 2.2., the authors describe that articles not meeting all four inclusion criteria were excluded. However, the four inclusion criteria were not reported. The authors should add them in this section.

Thanks for pointing this out. Inclusion criteria have been added on line 81.

- With regard to the literature search procedure, I was confused about the fact that even though only the search term “Acoustic* Cry” was used, also papers with variations of “cry” in the paper title were found, namely “crying” and “cries”. How can this be explained? Moreover, I wondered if more relevant articles would have been found, if specific acoustic terms would have been additionally used as search queries, such as “fundamental frequency”, “spectr*”, “cepstr*”, ...

- Section 3.1.: The authors state that the age of participants in cry studies ranges from 1 to 4 years with an average age of 133.5 days. Do the authors really mean 1 to 4 years rather than 0 to 3 years, i.e. the first to the fourth year of life?

Yes, indeed the age was reported in an incorrect way. The manuscript has been changed to reflect this. (Line 111)

-Could the authors please also present results on which single triggers were used for cry induction in the 70 studies that relied on non-spontaneous vocalisations?

Thanks for the point. This as been added in 3.1 (Line 124). The Data repository has been updated with the most recent version of the analysis notebook to reflect this change.

- At Checkbox 2 on Participant information in Appendix B, the authors should add that also the number of samples per participant has ideally to be reported in order to avoid that a group is misinterpreted by reason of a high number of samples by a single participant within the group.

This has been now added in B.2 (Line 506).

- At the Checkbox on “Region of Interest” in Appendix B, the authors should add the term “frequency” before “region of interest” in the explanation (frequency region of interest of the signal …).

Fixed (Line 536)

- Appendix A.2.3.: The authors should add that voice samples of interest recorded in home settings are often overlaid by background noise, such as everyday noise sources or parental voice.

Thanks for highlight this. It has been added  to the manuscript (Line 373)

- Appendix A.2.4.: How far does the sampling rate of a signal influence the lowest detectable frequency?

Following this, we noticed that lowest detectable frequency should have had been replaced with frequency resolution. This has been done in Added (Line 395)

- The authors should specify (in the Figure legend) the source-microphone distance used for measuring the microphone frequency response shown in Figure A1 (usually 1m?).

Figure A1 is a mockup of a possible microphone. Nevertheless, we added a note on the fact that microphones’ frequency response is evaluated at a specific distance and reported by the vendors.

Reviewer 2 Report

The present work is based on a very interesting topic, since the acoustic characteristic of infants’ cry and their implications on health can bring to innovative solutions for the prevention of diseases at older ages.

The general structure of the manuscript and the writing of it are very well done. Sections and paragraphs are well organized, and the reading of it is fluent and clear.

I propose the authors to make some minor changes to punctual parts of the manuscript.

- Introduction: the first lines of this section are effective and clearly introduce the aim of the work. However, from line 25 on some specific information about the contents of the revised papers are given, without a previous more generical introduction that I could find in the last paragraphs (lines 38-56). Therefore, I would suggest moving such paragraphs before and then giving the insights that are more specifically related to the papers’ contents. In such a way, the reader will be brought from a generical overview of the problem/aim to focused open questions that are deepened in the next sections.

- Conclusions: I would appreciate that the authors could formulate a list of lacks in the available literature highlighting for the different issues analysed in the “Results” section (i.e., participants, data collection, methods and data analysis) what can be considered as most effective, useful, accurate and replicable for future studies. As an example, which is the best equipment to be used? Which is the optimal number of monitorings to be carried out? And why? Giving personal indications on this will allow future researchers replicating experiments and choosing the most suitable procedures.

- Appendix A:

A1 > are there information about the presence/absence of known people close to the involved infants? The presence of parents or relatives, in fact, may influence the outcome of cry but it is not clear if such variable is ever considered in the studies or not. Then, together with the issue related to effects of the gender of cry, is there any information on the effects of ethnicity? Of course young infants still have an incomplete development of speech/language, but maybe there are physiological features that can be recognized even at such a young age.

A2 . A.2.2 > is there any study in which researchers make use of contact microphones instead of air mics? This would avoid the effect of environment on the recordings, and would allow monitoring the real vocal folds activity while crying instead of the acoustic emission that comprises the filter of the vocal tract and mouth too. Contact microphones are non-invasive and widely used in the research related to professional voice use (e.g., for teachers, singers, actors, etc), and provide accurate results.

A2 . A2.3 > are there information about air temperature and relative humidity in the considered studies? Such environmental features may have influence on the acoustic features of cry.

Author Response

- Introduction: the first lines of this section are effective and clearly introduce the aim of the work. However, from line 25 on some specific information about the contents of the revised papers are given, without a previous more generical introduction that I could find in the last paragraphs (lines 38-56). Therefore, I would suggest moving such paragraphs before and then giving the insights that are more specifically related to the papers’ contents. In such a way, the reader will be brought from a generical overview of the problem/aim to focused open questions that are deepened in the next sections.

Thanks for your input. We followed your input and the order of the paragraphs has been changed to enhance the readability of the introduction.

- Conclusions: I would appreciate that the authors could formulate a list of lacks in the available literature highlighting for the different issues analysed in the “Results” section (i.e., participants, data collection, methods and data analysis) what can be considered as most effective, useful, accurate and replicable for future studies. As an example, which is the best equipment to be used? Which is the optimal number of monitorings to be carried out? And why? Giving personal indications on this will allow future researchers replicating experiments and choosing the most suitable procedures.

As per this, we added a paragraph in 5 (Line 245). We tried to give some hints, whenever possible, while leaving some of the decisions to other researchers, as the adoption of this versus the other methods should be based on the specific needs of the research and variables under investigations.

- Appendix A:

A1 > are there information about the presence/absence of known people close to the involved infants? The presence of parents or relatives, in fact, may influence the outcome of cry but it is not clear if such variable is ever considered in the studies or not. Then, together with the issue related to effects of the gender of cry, is there any information on the effects of ethnicity? Of course young infants still have an incomplete development of speech/language, but maybe there are physiological features that can be recognized even at such a young age.

Unfortunately the articles are not always clear when it comes to the presence of known people, but we can hardly imagine studies conducted without the presence of at least a caregiver. A note on this has been added in Appendix B. For what concerns the ethnicity, there are not really study connected to this but there are some preliminary insights on the influence of caregivers’ language on infants’ vocalizations. Changes has been made to add a new section in A.1.8 (line 325) and in B (Line 519).

A2 . A.2.2 > is there any study in which researchers make use of contact microphones instead of air mics? This would avoid the effect of environment on the recordings, and would allow monitoring the real vocal folds activity while crying instead of the acoustic emission that comprises the filter of the vocal tract and mouth too. Contact microphones are non-invasive and widely used in the research related to professional voice use (e.g., for teachers, singers, actors, etc), and provide accurate results.

Not in our reviewed articles, but we agree this may help solving specific issues in the recording of cry samples. A note on this as been added in A2.1 (Line 346)

A2 . A2.3 > are there information about air temperature and relative humidity in the considered studies? Such environmental features may have influence on the acoustic features of cry.

Added both in A2 and B - Data collection (Line 383 and 525)